# Health and Safety Risk Mitigation among Artisanal and Small-Scale Gold Miners in Zimbabwe

**DOI:** 10.3390/ijerph192114352

**Published:** 2022-11-02

**Authors:** Josephine Singo, Dingani Moyo, John Bosco Isunju, Stephan Bose-O’Reilly, Nadine Steckling-Muschack, Jana Becker, Antony Mamuse

**Affiliations:** 1Centre for International Health, University Hospital, LMU Munich, Ziemssenstr. 5, 80336 Munich, Germany; 2Devsol Consulting, Clock Tower, Kampala P.O. Box 73201, Uganda; 3Exceed Institute of Safety Management and Technology, Kampala P.O. Box 72212, Uganda; 4School of Public Health, University of the Witwatersrand, Private Bag 3, WITS, Johannesburg 2050, South Africa; 5Faculty of Medicine, National University of Science and Technology, Ascot, Bulawayo P.O. Box AC 939, Zimbabwe; 6Faculty of Medicine, Midlands State University, Private Bag 9055, 263, Senga Road, Gweru P.O. Box 9055, Zimbabwe; 7Disease Control and Environmental Health Department, School of Public Health, Makerere University, Kampala P.O. Box 7072, Uganda; 8Institute and Clinic for Occupational, Social, and Environmental Medicine, University Hospital, LMU Munich, Ziemssenstr. 5, 80336 Munich, Germany; 9Department of Public Health, Health Services Research and Health Technology Assessment, UMIT-University for Health Sciences, Medical Informatics and Technology, Eduard-Wallnoefer-Zentrum 1, 6060 Hall in Tirol, Austria; 10Klinikum Osnabrueck GmbH, Am Finkenhuegel 1, 49076 Osnabrueck, Germany; 11German Professional Association of Private Practitioners in Hematology and Medical Oncology, Sachsenring 57, 50677 Cologne, Germany; 12Department of Geosciences, Midlands State University, Private Bag 9055, Senga Road, Gweru, Zimbabwe

**Keywords:** artisanal and small-scale gold mining (ASGM), artisanal and small-scale mining (ASM), Zimbabwe, multi-causal analysis, health, safety, multi-stakeholder risk mitigation, large-scale and small-scale mining collaboration, community and public health interventions, mitigation measures

## Abstract

Artisanal and small-scale gold mining (ASGM) is often associated with no or compromised attention to health and safety. Although headlines of fatal accidents in Zimbabwe characterise ASGM, little attention is paid to prevention strategies. This study, therefore, explores health and safety risk mitigation in ASGM in Zimbabwe to inform prevention strategies. A qualitative design was used with focus group discussions and in-depth interviews. Data were analysed using thematic analysis, coding, and descriptive statistics. Reported factors contributing to compromised health and safety included immediate causes, workplace factors, ASM related factors, and contextual factors, with interconnectedness between the causal factors. In addition, factors related to ASGM were significant. For risk mitigation, formalisation, organisation of risk reduction, behaviour change, and enforcement of prevention strategies is proposed. A multi-causal analysis is recommended for risk assessment and accident investigation. A multi-stakeholder approach could be considered for risk mitigation including community and public health interventions. However, risk mitigation has been characterised by gaps and weaknesses such as lacking ASM policy, lack of capital, poor enforcement, negative perceptions, and non-compliance. Therefore, we recommend addressing the threats associated with health and safety mitigation to ensure health and safety protection in ASGM.

## 1. Introduction

Mining is a hazardous occupation, accounting for 1% of the global workforce and 8% of fatal occupational accidents, with more people employed in artisanal and small-scale mining (ASM) than large-scale mining (LSM) [1]. In Zimbabwe, the zero-harm policy [2] targets the formal sector, which includes LSM and excludes the informal sector, e.g., ASM. Artisanal and small-scale gold mining (ASGM) is a branch of ASM that extracts gold. This study refers to both ASM and ASGM. The number of people working in ASGM worldwide is estimated at 14–19 million [3]. In Zimbabwe, more than 500,000 people are informally employed in ASM [4], mainly in ASGM. Furthermore, ASGM is often informal, undercapitalized, and characterised by little or no mining knowledge and limited capacity to comply with international or national health and safety regulations [1,5,6]. As a result, the number of occupational accidents in ASGM worldwide is six to seven times higher than in LSM [1].

In 2020, 5000 deaths from ASM were reported worldwide, while Zimbabwe had 42 deaths, based on an online media data set [5]. In addition, our previous articles revealed reported occurrences of health and safety problems (45%) [7], accidents (35%) and injuries (26%) [8]. The health and safety problems included respiratory (n = 33, 26.6%), musculoskeletal (n = 29, 23%), stress (n = 28, 23%), hearing (n = 11, 8%), and reproductive (n = 4, 3%) challenges [7] with little or no access to health care services [7,8,9,10]. Occupational injuries and health risks in mining negatively impact miners, their families, and ASGM communities [5,6,7,8,9,10,11]. The range of control measures from most protective to least protective consist of elimination, substitution, engineering, administrative controls, and PPE [12]. While most fatalities in ASM are preventable [6], preventive measures such as engineering controls, accident investigations, and consistent risk management are scarce in ASGM [7,8,13,14,15].

Fatalities are preventable if the causes of serious accidents, minor accidents, and near misses are investigated and managed [16]. Accident investigation involves determining the cause of an accident through risk assessment, followed by enforcement of measures to prevent future accidents [17]. Accident investigation has resulted in a significant reduction in accidents and fatalities in LSM [17]. Reason (2016) developed the Swiss cheese model for organisational accident risk management, which illustrates the causal factors of accidents (in high-technology systems) as unsafe acts, local workplace factors, and organisational factors [18]. Although Reason’s model (2016) is popular for identifying causes of accidents in manufacturing, it was found to be inadequate for different workplace settings [19]. Therefore, Bonsu (2017) modified the Swiss cheese model to include barrier analysis (existing control measures), meta-analysis (other factors), and causal analysis and applied the model to the study of mining accidents. [19]. However, ASGM is often associated with little or no consideration for health and safety [1,5,6,20]. Barriers to risk reduction in ASGM include ignorance, individual personalities, inherent informal practices, and financial constraints [5,6,15,21,22,23,24]. Figure 1 shows the contributing factors to hazards in ASGM and the actions that could mitigate health and safety risks in ASGM.

Figure 1a illustrates the immediate and underlying causes of compromised health and safety in ASGM. Figure 1b suggests mitigation measures including formalisation (incorporating financing and regulation of the sector), health and safety organisation, behaviour change, and enforcement through multi-stakeholder synergy including LSM-ASM public health and community collaborations and interventions. Proposed mitigation layers are dependent on human effort [12], hence each layer is characterised by numerous weaknesses symbolised by ‘holes’ as in the Swiss Cheese model [25], gaps such as the lack of resources, deficiencies in ASM policies, poor enforcement, unsafe equipment, hazardous work environment, poor supervision, and negative perceptions [5,15,21,24,26]. According to Reason (2000), the ‘holes’ in protective layers open and close in response to prevailing circumstances [25]. Therefore, efforts should be made to minimise gaps and weaknesses in each defence layer to protect health and safety in ASGM [8]. A single protective layer could increase the likelihood of accidents, injuries, and poor health [7,8], while an additional protective layer may improve protection of miners in ASGM [8]; see Appendix A. Appendix A [8] assumes significant reduction of deficiencies through successive and effective control measures during health and safety mitigation in ASGM [8].

Immediate causes are defined as unsafe actions and human error [18]. In addition, Bonsu (2017) included non-human causes such as equipment failure [19] in the Swiss cheese model. Therefore, in this study, human errors such as mistakes, non-compliance, human behaviour, and non-human causes (i.e., equipment) are defined as immediate causes.

Workplace risk factors are workplace conditions that can cause accidents [18], e.g., incompetence, inappropriate equipment, unsafe work practices, hazardous work environments, and noise levels that lead to accidents [18,19].

ASGM-related factors are the inherent causal factors associated with ASGM. For example, informal practices prevalent in the ASM sector, such as lack of regulation, lack of financing, low levels of education, insufficient capital, and lack of health and safety management [1,5,6,11,13,21], could contribute to deficiencies in best practices in health and safety mitigation. In addition, the Nigerian ASM sector’s lack of adequate information has been identified [23]. These shortcomings can thus contribute to accidents and poor health in the sector.

Contextual factors include those surrounding the incident, such as time of day, task, and seasonal changes, as well as external factors, such as the COVID-19 pandemic, also known as ‘metadata’ [19].

Comprehensive control measures are not standard in ASM [6,14,21,22]. An analysis of hazards and controls in ASGM revealed a lack of engineering controls as well as weaknesses in risk mitigation, including compromised use of PPE [7]. Compromised and inadequate control measures could lead to accidents, injuries, and ill-health in ASGM [6,7,8,11,24].

Risk assessment should be followed by risk mitigation, and a lack of effective risk mitigation leads to serious workplace accidents [17,27,28]. The plan-do-check-act (PDCA) method, which focuses on continuous improvement of health and safety, has been applied in the formal sector (including LSM in Zimbabwe [28]), which is characterised by conventional management systems [27,28]. In contrast, prevailing informal conditions in ASM, such as lack of financing and health and safety management [6,13,29,30], may challenge the applicability of the PDCA in ASGM.

Scholars have argued that ASM’s lack of regulatory oversight [30,31] hinders risk mitigation [6,13,21] in the sector. Therefore, there is the need to formalise the sector (Figure 1b) by adapting ASM policies, regulating and financing the sector [6,29,31]. The organisation of health and safety management (Figure 1b) incorporates defining roles, planning, and implementing health and safety risk mitigation measures (e.g., risk assessment, risk mitigation and training [27,32].

In addition, ASGM is characterised by negative workplace behaviour that leads to unsafe actions such as mining shaft pillars, non-compliance with the recommended post- blasting waiting period, and poor compliance with PPE use [6,13,24]. Behaviour is therefore a key predictor of workplace safety and health [24,25,28,33]. Figure 1b suggests behaviour change among risk reduction measures for ASGM. The behaviour change wheel identifies capability, opportunity, and motivation as prerequisites for behaviour change [34]. At the same time, safety behaviour change is influenced by workplace organisation, context, and existing systems [33]. In Nigeria, positive health and safety information-seeking behaviour was found in ASM [23]. In addition, training on health and safety was associated with best health and safety practices [22] in Ghana. Thus, despite unsafe actions associated with human error, Reason (2017) has defined humans as heroes who can adapt to situations to promote safety [35]. Safety culture is therefore described as a behavioural approach [6].

Implementation (e.g., ASM regulations, financing of health and safety in ASM, training on health and safety practices in ASM) and follow-up of health and safety practices could be promoted through a multi-stakeholder approach (Figure 1b) among the miner(s), mine owners, ASM associations, the Ministry of Mines and Mine Development, financial institutions, and other stakeholders including Environmental Management Agency (EMA). A ’cross-scale’ and ‘cross-level’ approach has been proposed to mitigate challenges in ASGM [36]. Hence, the need to integrate government agencies, local government representatives, community leaders, LSM, and public health programs in ASM risk management [6,11,21]. Furthermore, the literature has found LSM-ASM collaboration (Figure 1b) as one way to introduce health and safety in ASM [6,21,26]. The International Labour Organisation(ILO) (1999) and Smith et al. (2016) identified the need for a public and community health approach (Figure 1b) to mitigate health issues in ASM [6,11]. Therefore, a multi-stakeholder approach is suggested to reduce health and safety risks in ASGM (Figure 1b). In addition, poor sanitation and hygiene, malaria, tuberculosis, sexually transmitted diseases, and malnutrition are prevalent in ASM [6,7,9,10,11] highlighting the relevance of public and community health interventions [6,11] (Figure 1b), including awareness raising, counselling, screening, and outreaches to ASM communities.

ASGM is associated with hazardous working conditions, high exposure to dust, chemicals (mercury and cyanide), noise, vibration, poor ventilation, over-exertion, confined workplaces, inadequate equipment, and compromised safety practices, resulting in fatalities, respiratory diseases, and poor health [3,6,9,24,26]. The Minamata Convention on Mercury in ASGM, therefore, focuses on reducing and eliminating mercury use through initiatives such as the Planet Gold interventions [37]. In addition, international Non-Governmental Organisations (NGOs) have extended due diligence to ASM through programs such as the CRAFT Code [38]. However, a review of health and safety in ASGM reveals a lack of comprehensive health and safety management in ASGM [14], and the literature has therefore recognised the need for health and safety risk mitigation and prevention measures in ASM [11,13,21,24]. Furthermore, accidents and fatalities in ASM [39] usually assign the sector with blame rather than attempting to understand the causal factors or implementing mitigation measures [5]. Therefore, preventive health and safety programs in mining must be a global priority [40]. Yet, there is little research on risk mitigation in ASM. Thus, this study explores health and safety risk mitigation in ASGM to inform relevant interventions. The objectives include (1) exploring the causal factors for accidents and poor health in ASGM and (2) identifying an approach to risk mitigation for health and safety in ASGM in Zimbabwe.

## 2. Materials and Methods

Data were collected in 2017 and 2020 in Kadoma and Shurugwi in Mashonaland West and Midlands Provinces, which are characterised by high ASGM activity [41]. Smith et al. (2016) proposed the application of quantitative and qualitative methods to understand the nature of health and safety risks in ASM [11]. Hence, a qualitative research design was used to answer questions about the “what”, “how”, or “why” of incidents [42] to explore the factors and attitudes contributing to health and safety in ASGM in Zimbabwe. Data were also supplemented with data from focus group discussions (FGDs) and in-depth interviews (IDIs) from a previous study conducted in 2017 [10].

### 2.1. Focus Group Discussions 2020 Survey

Focus group discussions (FGDs) explored the factors contributing to health and safety challenges in ASGM in Zimbabwe. The focus group guide addressed issues such as accidents and injuries, health and safety measures, and practices (Appendix B). The focus group guide was developed, piloted at a site in Kadoma, and translated into Ndebele and Shona. The Medical Research Council of Zimbabwe (MRCZ) approved the research protocol and focus group guide. The FGDs were conducted at ASGM sites during working hours as part of a cross-sectional survey on health and safety in ASGM in Zimbabwe. The target population was artisanal and small-scale gold miners working in Kadoma and Shurugwi.

The 2020 survey targeted mining areas in Kadoma and Shurugwi where rudimentary and more mechanised mining methods were used [7]. A simple stratified random sample was then drawn by reshuffling the names of identified practising rudimentary and more mechanised mining methods [7]. Thirty-four sites were visited in Patchway, Battlefields, Sanyati, Mayflower, Brompton, and Mudzengi in Kadoma, and Wonderer and Chachacha in Shurugwi in the 2020 survey [8]. Participants for focus group discussions were then selected randomly from the visited sites. Consenting adults aged 18 years and older who had been engaged in ASGM for at least six months were included. Drunk, disinterested, and uncooperative respondents were excluded. The FGDs were conducted during the rainy season in November 2020 at mining sites at a reasonable distance from the sources of noise and destruction. FGDs from the 2020 survey consisted of 5–16 participants and were conducted in the participants’ language, Shona; all participants had Shona as their first language (there were no Ndebele speaking participants among the study sample). The sessions were recorded using a Samsung digital voice recorder, transcribed in MS Word in Shona, and translated to English and reviewed by the translator.

### 2.2. Data from Focus Group Discussions and In-Depth Interviews 2017 Survey

Transcripts of IDIs and FGDs, and summary notes of IDIs from another study conducted by the University Hospital of LMU Munich, Germany, in October 2017 [10], were used to supplement the findings. Snowball sampling was used in 2017 due to difficulties in accessing miners. Specifically: local partners were contacted first, and initial participants were approached for different sites. This process was continued. Eight sites were visited in Pingo, Patchway, and Lasgos [10]. FGDs focused on accidents and injuries, health and safety challenges, and existing practices. The topics for the IDIs targeted existing health and safety, the use of PPE, and related difficulties (Appendix C). The FGDs and IDIs targeted miners working in ASGM in Kadoma. The IDIs were conducted during working hours on mining sites and were selectively recorded at locations where the noise level was tolerable.

All study participants had Shona as their first language. The participants’ preferred language, English or Shona, was used for the IDIs. The principal researcher was a foreigner, and the participants’ preference for English facilitated direct interaction between the principal researcher and the study population. The FGD with the mine owners was conducted in an office in the town of Kadoma. The FGDs with miners took place on-site, at a reasonable distance from the sources of noise and destruction. The primary language of the participants (Shona) was used for the FGDs; FGDs were recorded. Summary notes were taken in Shona for the IDIs conducted by the research assistant who had Shona as a first language. IDIs which were conducted by the principal researcher in English were recorded. The FGDs and selected IDIs were transcribed in MS Word in Shona (FGDs) and English (IDIs). Finally, the transcripts and summary notes in Shona were translated into English and reviewed by a translator.

### 2.3. Data Analysis

The multi-causal analysis and multi-stakeholder risk mitigation framework (Figure 1) was used for data analysis. Transcripts of IDIs, FGDs, and summaries of the qualitative interviews were printed out and reviewed to familiarise the data. The data sets were imported into NVivo. Initial content analysis was performed through auto-coding in NVivo to visualise and understand the data. Coding was done using MS Word [43] with reference to the printouts reviewed, as access to qualitative research software was limited. NVivo was accessible during a short training session while the principal researcher for the 2020 survey had to perform the data analysis as part of the study. Emerging themes were identified. An initial classification of codes was created for recurring themes. Sub-coding was done according to the emerging sub-themes. Parent, child, and child–child nodes were identified and collated according to the conceptual framework (Figure 1). Mind mapping was done to visualise the codes compiled, as presented below. The frequencies of similar codes were aggregated as shown below. The data were quantified, and percentages were calculated in Microsoft Excel and analysed using descriptive statistics. The findings were presented in tables, graphs, charts, and direct quotations.

### 2.4. Ethical Approval

This work was approved by the University of Munich (LMU) Ethics Committee (Project 20-068) and the Medical Research Council of Zimbabwe (MRCZ/A/2603). Consent was sought from local authorities, mine owners, and all participants. Participation was voluntary and the informed consent form was signed before data collection in 2020 and 2017. The survey conducted in Kadoma in 2017 in the dry season was approved by MRCZ (MRCZ/B/1425) [10] and the University of Munich (LMU) Ethics Committee (17-665) [10].

## 3. Results

Three FGDs were conducted in 2020 with male miners (16 miners 45 min), female miners (5 miners, 21 min) and women who were living in a mining compound (15 women, 60 min). The focus group discussions were conducted on mining sites that were practising more mechanised mining methods in Shurugwi. Focus group discussions with men and women were conducted on the mining sites during working hours. Focus group discussions with women living in a mining compound were conducted on the mining compound at an appointed day and time. FGD were conducted on registered sites which were owned by individuals.

The findings were supplemented with existing data from three FGDs and 84 IDIs from the 2017 survey. Focus group discussions with miners were conducted on a registered mining site owned by an individual and a processing centre owned by an ASM mining company. The focus group discussion at the processing centre consisted of 17 miners (16 men and 1 woman) and was conducted for 58 min. The focus group discussion which was conducted on a mining site had 31 participants (men), participants were split into two groups of 15 (30 min) and 16 (28 min). Focus group discussions with mine owners (18 men, 65 min) were conducted in an office in Kadoma. Participants in the IDIs were aged between 19–70 years. IDIs had a response rate of 98%. Of the 84 IDI participants, 15 (18%) were women, and 69 (82%) were men.

Six transcripts of FGDs from the 2017 and 2020 surveys and seven IDIs transcripts from the 2017 survey were analysed, along with summary notes from 77 IDIs using the multi-causal analysis and risk mitigation framework, Figure 1. During the same study in the 2020 survey, the reported prevalence of health and safety problems was 45% (178) [7], accidents and injuries were reported as 35% (140) and 26% (103), respectively [8].

### 3.1. Multi-Causal Analysis of Risk Factors Contributing to Health and Safety among Artisanal and Small-Scale Gold Miners in Kadoma and Shurugwi, Zimbabwe, in 2017 and 2020

The identified causal factors included immediate causes, workplace factors ASGM-related factors, and contextual factors with no clear-cut distinction beteen the majority of the factors. Workplace factors such as unsafe shafts and incompetent people were found. The reported factors contributing to health and safety in ASGM are illustrated in Figure 2, below.

Underlying informal practices in ASGM influenced the workplace and immediate causes. For example, inherent informal practices and lack of training in the sector which in turn influenced unsafe work practices such as mining of support pillars.

Human behaviour included a negative perception of PPE use. In addition, non-compliance with known safety standards was widespread, as further presented below, Table 1; see Appendix A for more detail.

The rainy season was associated with the weakening of the shaft walls and shaft collapses due to wet ground conditions which confirms findings from the 2020 survey [7]. During the same survey (2020), rising unemployment and the declining economy, exacerbated by COVID-19, made ASGM a promising source of employment [7].

Findings based on the voices and direct experiences of miners are presented below.

#### 3.1.1. Workplace Risk Factors

The themes that recurred as workplace risk factors included environmental pollution, unsafe work practices, incompetent people, and psychosocial factors. For example, unsafe practices included overwork, lack of safety talks and checks, overcrowded shafts, and uncontrolled blasting. Psychosocial risk factors included workplace violence, stress, and family absence (Figure 2 and Table 1).

##### Environmental Chemical Pollution

Exposure to chemicals was the most common environmental hazard in the workplace. Miners were exposed to mercury, chemical dust, and toxic gases from blasting (Table 1). The retort with an enclosed protective system for gold-mercury amalgam burning had been introduced to avoid open amalgam burning and reduce exposure to mercury gas. However, there was low response uptake of the retort and open amalgam burning was common, Figure 3b. Cyanide leaching from mercury-containing tailings was practised, as shown in Figure 3a below.

Environmental mercury and cyanide pollution was common, Figure 3a,b. In addition, there was a site owned by a mining company that had an upgraded treatment facility for cyanide leaching from mercury-containing tailings (presented below), which is associated with increased releases of complex mercury and cyanide chemicals into the environment, with potential adverse impacts on the health of miners, their families, and ASGM communities. Open amalgam burning was practised (Figure 3b), and the use of mercury was preferred as shown below. Hammer mills were common in Shurugwi; it was typical for a processing centre to operate more than five diesel-powered hammer mills at the same time, which was associated with fuel emissions, noise, cramped working spaces, and dust from mercury-containing tailings (especially in the dry season), Figure 3c. Dust exposure was a predominant workplace risk factor (especially in the dry season) from dry drilling, milling, and tailings [7]. Furthermore, there was no safe management of chemicals as shown below. In 2020, during the same study, hazard identification and risk assessment identified numerous hazards in ASGM including chemicals, mine-contaminated drinking water, chemical contamination of farmland, and poor management of mine waste, and silica dust, which required attention [7]. Cigarette smoking at mining sites was also widespread during the 2020 survey. In addition, participants confirmed occurrences of tuberculosis (TB) among the miners in ASGM:


*‘When drilling, there is a need to cover the mouth and nose because dust from drilling can affect health, like the lungs and TB.’*
(FGD, men, Kadoma)

Miners were also aware of free TB treatment:


*‘With TB, you do not pay … but … You pay until diagnosed TB positive; then, you will be declared a government property (for free TB treatment).’*
(FGD, mine owners, Kadoma)

However, the miners could not afford the initial consultation and TB tests required for free TB treatment, hence the need for community and public health interventions on TB screening.

##### Unsafe Shafts

Unsafe shafts led to rock falls and collapsing mines. High temperatures, poor ventilation, and toxic gases could be associated with confined workplaces and the lack of ventilation routes. Accumulation of gases from blasting was typical in underground shafts. As discussed below, one participant pointed out the need for gas-testing equipment to monitor gases in underground shafts. In addition, miners did not adhere to the recommended waiting period for re-entry after blasting.

##### Noise

High noise levels were typical in milling centres, especially stamp mills. Noise levels were not controlled. As discussed below, one FGD participant was concerned about the lack of equipment to monitor noise levels (Appendix A). Despite the high noise levels, especially at sites with multiple stamp mills operating simultaneously, hearing protection was not standard, as presented further below. Accidents and injuries in the milling centres (especially at stamp mills as presented below) were also associated with poor communication between operators, which was influenced by excessive noise levels [7]; hearing problems were reported during the same study in the 2020 survey [7].

##### Violence

Workplace violence was reported:


*‘There are issues like failing to understand each other because of money. Typically, when there is money, there is fighting, violence, attacks, and stabbing.’*
(FGD, mine owners, Kadoma)

Conflicts and misunderstandings about money often resulted in violence. There were also reports of violent attacks by perpetrators nicknamed ‘*maShurugwi*’ (people from Shurugwi). In addition, violence was associated with disputes over unregistered claims at the discovery of rich deposits. One participant expressed that the lack of security guards and site registers exposed workers to invasions and violence as presented below.

##### Stress

Stress was reported as shown in Appendix A. Stress was also associated with being away from families. Miners could not visit their families regularly because of the uncertainty of their income:


*‘Challenges with visiting our families because of erratic salary.’*
(FGD with a men, Shurugwi)

Miners’ wages were unstable. Miners worked on a share basis, involving percentage sharing after profitable production, which was rare. In addition, for the salaried miners (monthly salary), consistent earnings were scarce because of unprofitable production as well as unfair employers, as presented below.

##### Lack of Training

Incompetence and lack of experience were prevalent among blasters and electricians (Table 1 and Appendix A). One participant commented that inexperienced blasting was associated with injury as was confirmed in the 2020 survey during the same study [7].

#### 3.1.2. ASGM Sector-Related Factors

ASGM related factors (Table 1) included lack of capital, lack of PPE, lack of training, insufficient control measures, and informal practices. In addition, the lack of money could be associated with the lack of PPE and the inability to invest in effective control measures (Table 1).

##### Lack of Finance

Miners reported having no financial support. As a result, individual miners could not fund health and safety equipment.


*‘It is difficult to follow safety rules because some things require you to put money down when looking for cash. So, you can say “chero zvazvaita” (“whatever happens”) to get money to take care of the family instead of working for health and safety.’*
(FGD, male miners, Shurugwi)

Moreover, PPE was not affordable, for the miners engaged in subsistence mining:


*‘[PPE] is not expensive. The only challenge is that rent is also needed.’*
(FGD, with men, Kadoma)

Most miners had low incomes and family obligations. In 2020, 229 (61.2%) of the miners worked based on sharing a percentage of profits. Since returns were not always profitable, earnings were low and irregular [7]. One participant commented that a work suit could wear before the next payment.

Some mine owners who participated in the FGD in Kadoma were aware of safety measures. These included standard shaft support of the mine, the sinking of standard shafts with adequate working space, the provision of two shaft entrances for better ventilation, and an escape route for emergencies.


*‘Good ventilation is expensive to achieve. We do collective bargaining. People cannot work to develop the mine; they want to mine where there is gold. If shafts were to be adequately sunk, two ventilation shafts would be needed for good ventilation. We use…. survival tactics without proper mining, which causes unsafe working conditions.’*
(FGD, Kadoma mine owners)

Miners could not introduce effective control measures, such as standardised mine development.

One mine owner indicated a willingness to comply with health and safety standards:


*‘[The] willingness is there; what is needed is money.’*
(FGD with mine owners, Kadoma)

Despite the desire to comply with safety standards, miners had no capacity to pay for safety.

##### Lack of On-Site Security Guards

Most sites lacked security personnel, and miners expressed that the lack of on-site security exposed workers to violence, as noted above.


*‘Mines involving more extensive operations need security and a registry of the names of people working in a specific shaft because we have many issues. When there is no registry, perpetrators can get into a shaft; there are cases of looting that can involve injuries to the people working in the shaft. “MaShurugwi” (people who come from Shurugwi) perpetrators invade because there is no security to protect the workers.’*
(FGD with men, Shurugwi)

Miners were at risk of assault, attack, and injury without strong on-site security. At the same time, it was expensive for mine owners to pay for security services:


*‘The other challenge is security; security must prevent raiding, but security is costly. I have just received a bill for 2350 USD, which has been accumulating.’*
(FGD, male mine owners, Kadoma)

Security guards were essential to preventing violence and raids; however, miners were limited in their capacity to pay for professional security services.

Violence was also associated with informal mining activities when a mining licence was granted for unregistered claims occupied by informal miners as presented above. The registration process was expensive for informal miners.

##### Compromised and Missing Control Measures

Compromised and missing control measures occurred 37 times in the qualitative data (Appendix A), representing 25% of the ASM related factors. During the same research period, PPE was the most frequently cited control measure, and use of PPE was associated with gaps. Furthermore, effective engineering control measures were absent, such as safe mine support, wet blasting, and acceptable noise levels [7,8].

##### Informal Practices

These included gold rushes, lack of ASGM-specific regulations, lack of enforcement of existing regulations, and unregistered miners (Table 1 and Appendix A).

Gold rushes were associated with high migration, overcrowded shafts, uncontrolled blasting, weak walls, and rock falls:

*‘*[There were] *too many people in an old shaft, 5000 or more people working with no control, uncontrolled blasting, weak points, and rockfalls 48 or more levels.’*(migrating miner, male, 32 years ID1, Kadoma,2017)

The migrating miner stated at he did not have time to use PPE during gold rushes.

The reported informal practices in the sector led to violations of known practices such as the use of PPE and mercury use, the observance of post-blasting waiting periods, and the continuous unsafe use of mercury, as presented below.

##### Lack of Accident Investigation, Accident Reporting, and Access to Health Care

Accident investigations were conducted externally in the event of a fatal accident. However, mine owners stated that that mine collapses were associated with fear of the police:

*‘*[If a] *mine collapses, it is difficult to get help, and colleagues can run away because of the police. If the case is reported, the police will ask for a statement issue to be reported so the fellow miners keep quiet. The incident may go unreported, yet someone could be rescued and saved.’*(FGD, mine owners, 2017, Kadoma)

As a result, accidents were rarely reported, as confirmed in the 2020 survey [7,8], necessitating cooperation with rescue services.

In the same survey, access to health care was described as limited, and there was no access to health insurance nor social security coverage for ASGM [7,10].

#### 3.1.3. Immediate Causal Risk Factors

The factors that were consistently cited as directly contributing to accidents and injuries (Appendix A) were behavioural (58), non-compliance (violations of known standards) (16), and equipment (3), as illustrated below (Figure 4).

Violations consisted of non-compliance to known best practces Table 1. At the same time non-compliance and behavioural factors were dorminant as presented below.

##### Behavioural Risk Factors

The most significant behavioural factor was the negative perception of PPE and mercury use (Table 1). Some miners even believed that working without PPE was necessary:


*‘The issue is when hitting with the hammer, you hit hard and sweat, and it’s preferred for one to be free without PPE because one will be working hard.’*
(FGD, male miners, Kadoma, 2017)

This was confirmed by informal conversations, which revealed a preference to work in old and torn clothes known as “*pendenge*”. In addition, compliance with PPE use was found to be difficult for informal miners employed by registered miners and ASM mining companies which were providing PPE. Mine owners in FGD in Kadoma reported experiences of miners entering shafts wearing PPE and then removing it while working underground. The provision of PPE was also associated with the high turnover of informal miners, as informal miners tended to move from site to site after receiving PPE. Moreover, the Chinese-manufactured respirator was considered uncomfortable, while the comfortable respirator was said to be expensive. At the same time, non-compliance with PPE use while working underground was also associated with the cramped working space and high temperatures underground (Table 1 and Appendix A), as explained below.

##### Alcohol and Sexual Behaviour

Alcohol abuse in the workplace and sexual indulgence were among the immediate causes. Focus group discussions revealed concerns about the lack of free condom distribution in ASGM communities,


*‘STIs (sexually transmitted infections) like syphilis. There are no free condoms. Sex is without protection.’*
(FGD, men, Shurugwi)

Sex was unprotected, and sexually transmitted infections were common. In addition, miners were away from their families, and it was not easy for miners to visit their families, as explained above.

It was also reported that the spread of HIV was typical when miners had money:


*‘Then getting too much money harms people when they get over-excited and get involved in sexual indulgence …. we have lost colleagues to HIV.’*
(FGD with mine owners, Kadoma)

Therefore, community and public health interventions are needed in ASGM communities.

##### Non-Compliance and Violations

There were violations of the use of PPE and existing regulations, including standard shaft support, mine ventilation, escape routes, winch rope change, post-blast waiting time interference, and mining pillars, as was confirmed during the same study [8].

One participant confirmed informal practices that resulted in a violation of mining standards:

*‘*[When] *working underground, I open a small hole…with poor ventilation, and I use explosives in that tiny hole. Despite the fumes from the blasting, I immediately get into the small hole after blasting to get money faster to pay rent at home.’*(FGD, male miners, Kadoma)

The miners did not comply with the mining regulations on shaft sinking and post-blasting waiting period as confirmed in the 2020 survey [7].

The violations of mining standards were related to the cost of standard shafts and the scarcity of trees for recommended timbering:


*‘Shafts must be made safe. People must use cement for shattering, but the prices are high…., and people may decide to use timber. Trees for timbering are becoming fewer and fewer in the forests. Some types of trees that used to be there in the past cannot be found in the woods or are now in distant forests, and it is costly to get them. Most people, therefore, work without timbering, resulting in collapsing mines.’*
(FGD, mine owners, Kadoma)

In addition, the FGDs revealed that non-compliance with PPE use was due to the limited working space and poor ventilation underground:


*‘Most of the shafts have poor ventilation, below the expected standards. As a result, some miners say they cannot wear protective clothing; the problem is limited working space and poor ventilation. In the standard shafts, people can work with protective clothing.’*
(FGD, mine owners, Kadoma)

The discomfort of wearing PPE was also due to the limited working space underground.

Furthermore, some miners knowledgeable of the recommended post-blasting waiting period did not comply:


*‘… our job is risky, even with blasting, you must get in after a specific time after blasting, but with us, our work is “chikorokoza” “informal”, you get in just after blasting when the fumes are still there.’*
(FGD, male miners, Kadoma)

Although there were individual miners who were informed about the post-blasting waiting period, there was no compliance. There was also the case of a site operated by a Chinese company where workers were forced to re-enter shortly after blasting. The miners at the said site worked on a wage basis and were not paid for months despite being in production. Further discussions with the miners revealed that there was no grievance mechanism. Therefore, the only option for the informal miners was to migrate to a different site without compensation, which was quite common.

##### Equipment Use

Causal risk factors included using equipment without safeguards, no use of PPE, increased mercury and cyanide pollution, dust from dry crushing and dry milling (hammer mills), and unsafe electricity, as shown below.

As shown above (Figure 5a–f), some registered mining companies sought high returns by switching to ASM mechanised equipment. However, informal practices, individual behaviours, and lack of training in equipment operation, management, and maintenance could contribute to health and safety problems, as demonstrated by the lack of safety precautions (Figure 5d), failure to follow standard operating procedures for personal protection (Figure 5a–d), and unsafe management of electricity (Figure 5f). For example, in the 2017 survey one miner operating a ball mill without gloves confirmed that he had forgotten to wear gloves [7] while another miner working in contact with cyanide stated a tendency of forgetting to wear the gloves [10]. Nevertheless, the miners found working with gloves uncomfortable. As can be seen in Figure 5, some individuals and mining companies had mechanised equipment without control measures. Technological progress was also associated with a need for industrial power, and unsafe electrical wires (Figure 5f) posed a risk. At one site, the mining camp had unburied electrical lines and children were playing on the compound.

#### 3.1.4. Contextual Risk Factors

The rainy season and the general economy were the identified contextual factors in Table 1. The 2020 survey was conducted during the COVID-19 pandemic, which was characterised by inflation and rising unemployment. More than 50% of the respondents were between 18 and 35 years old [7]. Overworking (Table 1 and Appendix A) was one of the workplace factors that could be associated with long working hours and night shifts. In the 2020 survey, long working hours, and an age range between 18–35 years were associated with increased accidents and injuries [7]. Hence, the need to understanding contextual risk factors in risk mitigation in ASGM in Zimbabwe.

### 3.2. Health and Safety Risk Mitigation Practices Reported from Focus Group Discussions (FGDs) and In-Depth Interviews (IDIs) in Kadoma and Zimbabwe in 2017

Table 2 presents the potential health and safety mitigation measures indicated by miners.

While health and safety management practices are rare in ASGM, health and safety management opportunities were identified [8] among ASGM miners in Zimbabwe.

FGDs and IDIs revealed that ASG miners were aware of the need for appropriate ASM policies, funding, and regulation of ASM activities. One of the miners argued that many ASGM miners in Zimbabwe had mining licences. However, the miners had no capacity and no financial or technical support to develop a safe shaft for underground mining:


*‘Ventilation needs capital, mine development needs capital. Capital should be made available to miners after getting the licence.’*
(FGD, mine owners, Kadoma, 2017)

The mine owners acknowledged the need for financial support after acquiring the licence.

The mine owners pointed out the need to regulate PPE use to avoid incurring costs related to a worker’s injury. However, it was pointed out that formalisation could be resisted in the ASGM sector, if associated with payment of taxes.

Some of the miners interviewed indicated that supervisors and managers were available at their worksites. For example, a woman who owned an operation in Kadoma (2017) expressed that she had a supervisor and a blaster on her site and the supervisor was responsible for safety checks:


*‘We do mining. Before mining, the leader should go down to check on safety as well as on the timbering. If not safe, safety measures are taken, after which the miners can go down and work.’*
(FGD, female miner, FGD with male and female miners, Kadoma, 2017)

The FGD above was conducted at the processing site, however, such practices were not observed at the sites visited.

In 2017, two sites operated by different mining companies in Kadoma had health and safety management structures. For example, one of the two sites had an occupational safety and health (OSH) department and a health care unit, which is rare in ASGM. (Unfortunately, the site with an OSH department was closed during the 2020 survey due to the COVID-19 pandemic). In addition, one individual mine owner interviewed off-site in 2017 indicated that he provided and monitored the use of PPE at his site.

The manager from one site operated by a mining company indicated that the company was paying for workers’ health care costs including routine medical check-ups.

On the other hand, two registered ASM mining companies (one in Kadoma and one in Shurugwi) that operated processing centres and were working on a percentage sharing basis with informal, artisanal miners, whereby artisanal miners were supplying ore to the processing centres, the registered ASM companies were not providing for the safety needs of the informal miners. While the ASM companies were aware of safety practices, the informal miners in partnership with the ASM companies were not capable of observing any best practices.

The cost of medical check-ups was reported as prohibitive for the average miner, as described below:


*‘Check-ups are good; the problem is money. Consultation fees, x-rays, and blood tests are expensive.’*
(FGD mine owners, Kadoma, 2017)

While the miners acknowledged the relevance of routine bio-monitoring in the workplace, medical check-ups were not affordable for the average miners. Hence the need for community and public health interventions involving awareness raising and screening (e.g., HIV, TB, and malaria), and distribution of mosquito nets and condoms. For instance, one respondent from FGD with mine owners indicated that mosquito nets were distributed once in ASGM in Kadoma by the outreach department from Kadoma hospital (mosquito was identified under biological hazards, Appendix A). During the same study it was found that Kadoma hospital had limited resources [7].

A preference for the use of mercury was also noted. However, miners indicated that the uptake of mercury-free technology depended on the efficiency of the new technology:


*‘It depends on how the technology works. We are convinced after seeing how it works. At the moment we are convinced that we extract all the gold if we use mercury.’*
(FGD, male miners, Kadoma, 2017)

There was a possibility of acceptance of an efficient mercury-free technology comparable to the use of mercury. Furthermore, the mine owners pointed out that mercury was expensive:


*‘If you come up with another technology and demonstrate the gold recovery percentage, we could opt for that because mercury is also a cost to us.’*
(FGD, mine owners, Kadoma, 2017)

Behavioural change in the use of mercury could be achieved through efficient alternative strategies.

## 4. Discussion

The contributors reported as influencing health and safety in ASGM were workplace factors, ASGM-related factors, immediate causes, and contextual factors. Workplace and ASGM-related factors were predominant, as confirmed by previous studies [6,9,10,13,20,21,22]. Harmful behaviour, workplace violence, unsafe shafts, limited capital, lack of training, equipment, alcohol and drug use, inadequate risk control measures, and informal practices were found as documented in the literature [6,7,8,9,10,24,26]. Multi-causal analysis was applicable to understand causal factors associated with health and safety in ASGM in Zimbabwe. Risk mitigation measures were identified, including formalisation and organisation of health and safety risk reduction. The proposed risk mitigation framework illustrates gaps and weaknesses in each level of risk mitigation [25]. This section discusses inter-linkages between causal factors, health and safety risk factors associated with equipment, and risk mitigation measures characterised by threats.

Immediate causes, workplace factors, ASGM-related factors, and contextual factors were intertwined. As documented in the literature, behavioural factors were among the immediate factors [24,25,33]. However, the causes of accidents and injuries in ASGM sector were interrelated. For example, the practice of sinking unsafe shafts (workplace factor) was a violation of required and known best practices [44] (immediate cause), which was associated with a lack of capital (ASGM-related factor) and exacerbated by the overall economic situation (contextual factor). Furthermore, deforestation had led to a shortage of traditional timber for shaft support. At the same time, previous work has recognised that ASGM is associated with unsafe human behaviour [1,5,6,22,24,26]. Alcohol abuse, preference for the use of mercury over the retort, environmental pollution, and HIV were also reported. However, scholars have confirmed that accidents and poor health associated with individual behaviour in ASM are also due to underlying causes such as lack, ignorance, and the illegal nature of the sector [6,20,23,30,31]. For example, free condoms were not distributed among the ASGM sites visited in Shurugwi, and there were no accepted mercury-free technologies. TB was also reported with limited access to health care. Therefore, mitigation strategies must address behaviours and underlying causal factors [11,18]. Public and community health interventions [6,11], such as awareness-raising and HIV and TB screening and treatment, could also be considered for ASGM.

The use of equipment was one of the immediate factors resulting in accidents and injuries, as documented in the literature [5,8,45]. It was common for miners to use equipment without PPE, confirming previous findings [5,6,10,46]. In addition, blast operators had low levels of competence [6,45,46]. Existing literature shows that blasting and drilling in ASM is dependent on unskilled workers [6,26,47]. Blasting and drilling in sub-standard shafts with poor ventilation and lighting were challenging, as revealed in the literature [48]. It was also typical for blasters to drink traditional beer and take drugs while working, confirming previous studies [10,45,49]. During the 2020 survey, accidents and injuries were more likely to occur at ASGM sites that had transitioned to mechanised ASM equipment which involved underground mining and mechanised processing [8]. At the same time, alcohol and drug use at work is associated with increased injuries and accidents [6,49]. Furthermore, noise pollution from equipment also leads to poor communication and an increased risk of accidents [19]. Increased throughput from mechanised ASM processing plants increases mercury and cyanide complex pollution [36]. While equipment is associated with increased production, technical support on safety precautions and guidance on basic standard operating procedures are essential to mitigate safety risks for mechanised ASGM [45,48]. Yet this mitigation measure is compromised by the level of literacy [6], human behaviour [33], financing, and illegalities intrinsic to the sector [6,29,30,31].

Formalising the sector by adapting ASM policies, regulating ASM, and funding the sector [5,6,26,29,30] is a prerequisite for effective risk mitigation in ASGM [14,31]. Despite mine owners’ willingness to implement preventive safety measures, access to bank loans was limited, especially for miners without collateral [14,30,32]. As a result, miners had limited opportunities to invest in safety and informal practices were common. In addition, as substantiated in the literature, there was a tendency for gold rushes associated with the uncontrolled mining of support pillars, which led to accidents and injuries [6,31]. It was also prohibitively expensive for most miners to hire security guards, and their absence was linked to violence and looting. Violence and evictions were commonplace at unregistered sites when rich gold deposits were discovered [31]. Therefore, adaptive policies, sector regulation, and funding are critical mitigation measures [13,24,30,36]. The formalisation of the sector could lead to the organisation of risk mitigation and transformation of the ASM context, which could motivate behaviour change. For example, formalisation made it easier for ASM workers to join the government social security system and health insurance in Mongolia [50] and Bolivia [6]. However, formalisation also comes with challenges. For instance, ASM regulation without implementation is insufficient, as informality in ASM is also associated with the lack of inspection and enforcement, not lack of regulation [6].

Management is fundamental to risk mitigation [21,27]. The Zimbabwean mining regulation (the same for both LSM and ASM) requires the mine owner to appoint a mine manager [44]. Managers and supervisors were available, especially on registered sites owned by ASM mining companies. Mine owners, sponsors, and individual miners were also identified as responsible for health and safety management in the 2020 survey [8]. For the ASGM sites with management, health and safety risk mitigation could be integrated into the daily management routine. This integration was suggested for ASM companies in Rwanda, where health and safety management in ASM was developed with a focus on the ASM mining companies, and roles were defined according to the existing management system [51]. On the other hand, there was a nexus between registered ASM mining companies and informal artisanal miners [6]. Capable registered ASM companies could be among the stakeholders for implementing health and safety in ASGM in Zimbabwe. In Mongolia, OSH roles and responsibilities were defined among ASM partnerships [32]. However, ASM regulation is still under discussion in Zimbabwe and advocacy for it is ongoing and requires more support from policymakers [29,30,31]. The CRAFT Code [38] and the Planet Gold Criteria [37] assign site accountability for responsible mining standards to a volunteer, which may not address the accountability gap [21] outside the structured context of the Planet Gold Criteria and the CRAFT Code. Cross-scale and cross-level interventions have therefore been proposed for ASGM [21,36]. This raises the need for a multi-stakeholder approach, including LSM-ASM collaboration [6,21,26]. In Mongolia, formalisation resulted in the institutionalisation of the sector and the distribution of ASM health and safety roles and responsibilities among the relevant ministries and stakeholders [32]. Furthermore, the literature points to the support of large companies, including LSM, in implementing health and safety mitigation measures [6,11,21,36], which could be threatened by weak stakeholder engagement [21].

Vocational training in ASM mining can improve miners’ skills while creating the opportunity for economic benefits [5,6,22,26,36], thereby facilitating behaviour change. In Ghana, training on health and safety correlated positively with health and safety practices [22]. In Mongolia, vocational training was conducted in mining districts [32]. Training outreach into ASM communities could also improve participation of marginalised miners [36]. Furthermore, training on accident investigation, risk assessment, and risk mitigation could be incorporated in ongoing interventions such as mercury-free demonstration sites [37]. Jennings (2002) has proposed simple accident reporting conducted by independent agencies focusing on health and safety [52]. Financial institutions interested in the sector could make essential health and safety requirements a prerequisite for financial support to incentivise health and safety mitigation and motivate behavioural change [33]. In addition to training, miners could be encouraged to adhere to good practices in health and safety, for example, facilitating the acquisition of the necessary equipment, for instance access to PPE and shaft support material to promote behaviour change. However, training could be hampered by high staff turnover, the migratory nature of the sector, and low response uptake [6,8]. In the 2020 survey, the migration rate among miners was 28% [8]. In addition, miners expressed low uptake of the retort (after training) and expressed the need for efficient mercury-free technology, which has been documented in the literature [6].

Introducing health and safety measures for ASGM is complex and requires multiple approaches. The proposed health and safety protection layers are characterised by numerous gaps and weaknesses that may open or close as determined by prevailing workplace conditions [25], hence the need to identify and address the underlying causes and turn threats into opportunities for promoting health and safety in ASGM. Moreover, safety is context specific. For example, a site with a single compromised layer of PPE will be vulnerable to adverse health and safety impacts as discussed in our previous articles [7,8], (Appendix A) [8]. In contrast, a registered site that can invest in safety, e.g., a mining ASGM company with an OSH department (Appendix A), can protect the health and safety of its workers [8]. Thus, a multi-stakeholder approach that includes capable ASM companies is needed to protect the health of marginalised informal and disempowered ASGM miners.

Limitations: The findings of this study were self-reported; recall and response biases were possible, and findings should be interpreted with caution. In addition, IDIs were recorded selectively due to high noise levels, especially in the processing centres. The generalisability of the results may therefore be limited. However, the results can serve as a guide and source of information for relevant policies and interventions.

## 5. Conclusions

This study has uncovered numerous factors contributing to health and safety in ASGM in Zimbabwe, as well as weaknesses in mitigation measures. The intertwined factors contributing to accidents and injuries in ASGM were identified as immediate causes, workplace factors, ASGM-related factors, existing control measures, and contextual factors. Environmental factors in the workplace and illegal practices were dominant. A multi-causal analysis could be applied to identify the immediate and underlying causes to inform risk assessment and accident investigation. Risk mitigation involves successive layers, such as formalisation with a focus on ASM policy, funding and regulation of the sector, organisation of health and safety mitigation, behaviour change, enforcement, and monitoring. Mitigation requires a multi-stakeholder synergy, such as LSM-ASM collaboration, division of roles among relevant sectors, and public and community health interventions. Each mitigation layer has gaps and weaknesses. Hence the need to address threats to health and safety protection in ASGM. In addition, further operational research is needed on health and safety risk mitigation in ASGM. This article proposed health and safety risk reduction in ASGM. However, the informal sector e.g., ASGM is usually marginalised from health and safety prevention initiatives such as zero harm [2]. At the same time the informal sector including, ASGM employs a larger population (exposed to inherent hazards without effective health and safety prevention measures) than the informal sector [1], hence the need for future work and occupational safety and health to consider prevention initiatives such as zero harm for both the formal and informal sectors through addressing barriers to health and safety mitigation in the informal sector (ASGM) at the multi-stakeholder level, as discussed above.

## Figures and Tables

**Figure 1 ijerph-19-14352-f001:**
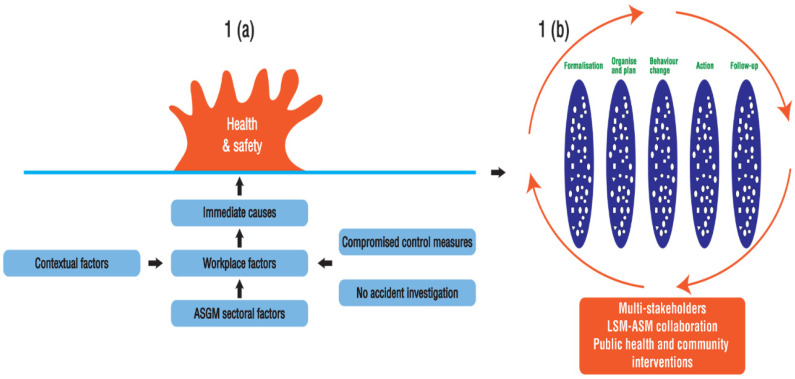
Multi-causal analysis and multi-stakeholder risk mitigation framework for health and safety in ASGM. (**a**) Multi-causal analysis to identify factors contributing to compromised health and safety in ASGM. (**b**) Multi-stakeholder risk mitigation framework for health and safety in ASGM. Source: Own.

**Figure 2 ijerph-19-14352-f002:**
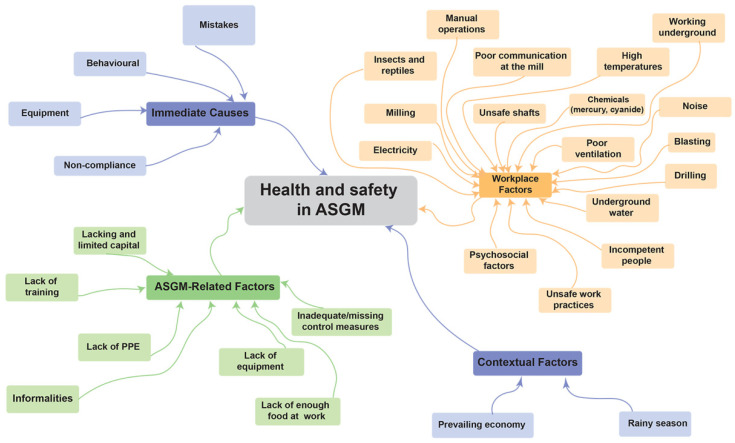
Mind map: multi-causal analysis of reported risk factors contributing to health and safety in ASGM in Kadoma and Shurugwi in Zimbabwe in 2017 and 2020.

**Figure 3 ijerph-19-14352-f003:**
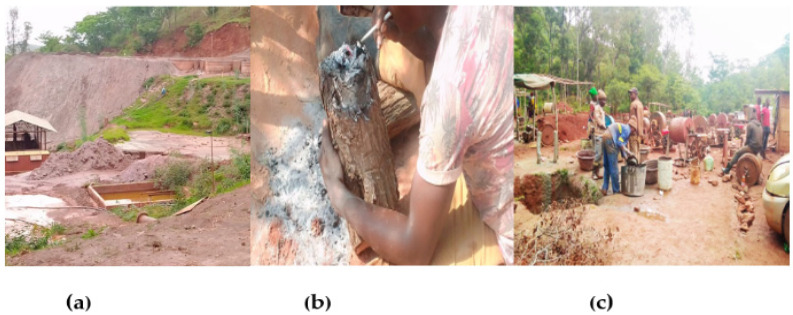
Chemical pollution in the workplace: (**a**) chemical contamination: cyanide leaching from mercury-containing tailings and dust from tailings, Shurugwi 2020; (**b**) open amalgam burning, Kadoma 2020; (**c**) fuel emissions from fuel-powered machinery and dust from mercury-containing tailings, Shurugwi 2020.

**Figure 4 ijerph-19-14352-f004:**
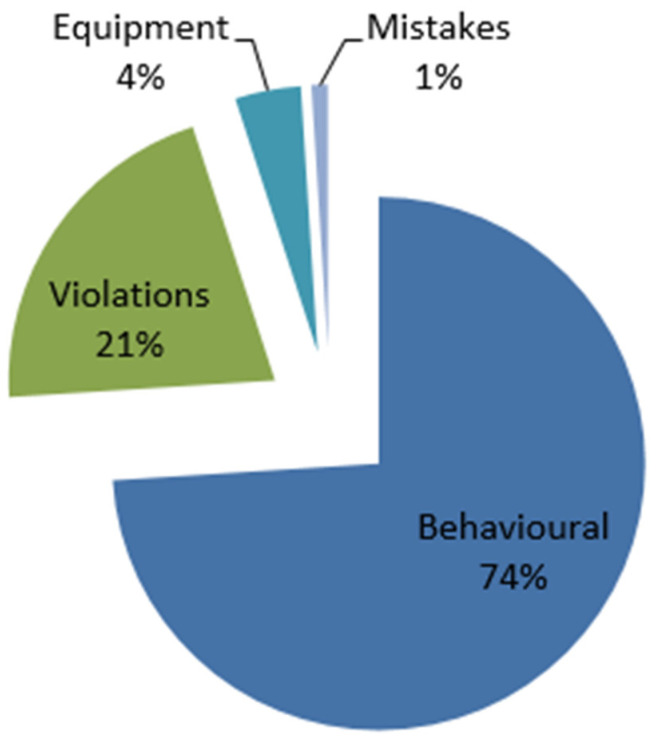
Distribution of immediate causal factors among artisanal and small-scale gold miners in Kadoma and Shurugwi in Zimbabwe in 2017 and 2020.

**Figure 5 ijerph-19-14352-f005:**
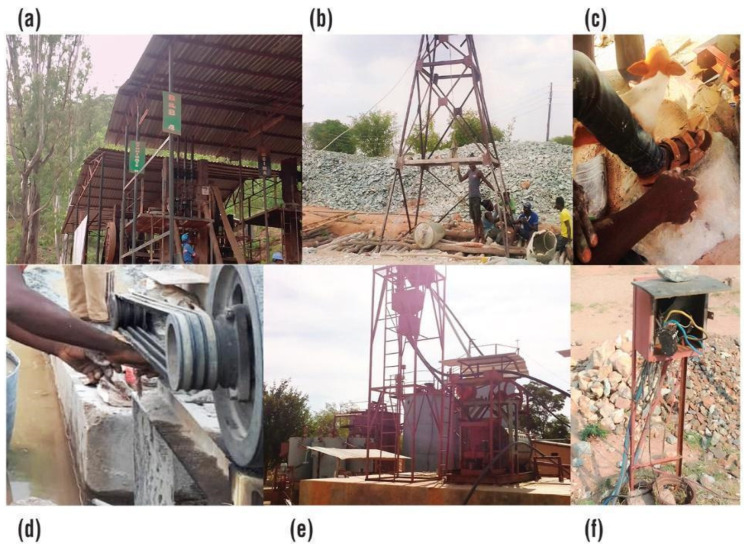
Causal risk factors associated with equipment used by artisanal and small-scale gold miners in Kadoma and Shurugwi in Zimbabwe in 2020: (**a**) hazardous noise exposure at a site with six stamp mills operating simultaneously with no technical measures to reduce noise and with no hearing protection; (**b**) mining site with improved equipment for underground mining, headgear is used instead of a winch to transport people and material, workers under the headgear do not wear PPE; (**c**) headgear operated without safety shoes, the operator wears open-toed shoes while operating the headgear with bare hands and without gloves; (**d**) hammer mill operator in contact with moving parts of the hammer with no safeguards, the operator wears no gloves; (**e**) mechanised ASGM processing plant capable of increasing the throughput of cyanide leaching from mercury-containing tailings at a mine site close to a mine compound, without environmental monitoring of chemical pollution; (**f**) unenclosed outdoor electricity junction box.

**Table 1 ijerph-19-14352-t001:** Multi-causal analysis of factors reported by artisanal and small-scale gold miners as posing safety and health risks in Kadoma and Shurugwi in Zimbabwe in 2017 and 2020. Numbers in parentheses illustrate the total recurrences of identified themes and sub-themes in the qualitative data. See Appendix A for more details.

Immediate Causes (78 Total Recurrences in Qualitative Data)			
Harmful Behaviour (58)	Negative perceptions (35)	Preference for mercury use (8)	Alcohol abuse (5)
	Unsafe sexual behaviour (1)	Workplace violence (9)	
Non-compliance (16)	Non-compliance, PPE (9)	1* Compromised waiting (3)	
	2* No development (2)	Mining survival tactics (1)	Mining of pillars (1)
Mistakes (1)	Forgetting to wear gloves (1)		
Equipment (3)			
**Workplace Factors (246 Total Recurrences in Qualitative Data)**			
3* Chemicals (103)	Chemical dust (34)	Mercury (29)	Toxic gases (25)
	Cyanide (8)	Acid and caustic soda (4)	
4* Unsafe shafts (34)	Rock falls (16)	Collapsing mines (8)	Sub-standard shafts (3)
	Slipping (1)		
Other workplace factors (109)	5* High temperatures (10)	Milling (8)	Biological hazards (8)
	Confined workplaces (3)	Poor sanitation (9)	Underground water (5)
	Blasting (4)	Poor ventilation (3)	Poor communication (2)
	Incompetent people (4)	Psychosocial factors (11)	Drilling (4)
	Noise (11)	Manual operations (2) Electricity (1)	Working Underground (2) Unsafe practices (22)
**ASGM-Related Factors (146 Total Recurrences in Qualitative Data)**			
Inconsistent income (7)	Lack of PPE (37)	Weak winch rope (2)	Lack of equipment (3)
Subsistence mining (2)	Limited capital (5)	Lack of security (1)	Lack of training (24)
Workplace starvation (8)	Compromised controls (37)	Informal practices (20)	
**Contextual Factors (3 Total Recurrences in Qualitative Data)**			
	Rainy season (2)	Prevailing economy (1)	

1* Compromised recommended waiting period after blasting; 2* Neglecting the initial stage of mine development; 3* Chemicals (n = 3) Total for all chemicals (n = 103); 4* Unsafe shafts (n = 6) Total for unsafe shafts (n = 34); 5* High temperatures underground shafts and summer time.

**Table 2 ijerph-19-14352-t002:** Emerging themes on health and safety risk mitigation measures reported by artisanal and small-scale gold miners in Kadoma in Zimbabwe in 2017.

Theme	Examples of Quotes from Focus Group Discussions (FGD)
Formalisation and financing	*‘There should be mine regulations for PPE to be checked at entry… and underground. Safety management should be available to monitor the use of PPE because if not monitored, it’s still a cost when a worker gets injured.’* (FGD, mine owners, Kadoma, 2017) *‘Ventilation needs capital, mine development needs capital. Capital should be made available to miners after getting the license.’* (FGD, mine owners, Kadoma, 2017)
Organise and plan	*‘We do mining. Before mining, the leader should go down to check on safety as well as on the timbering. If not safe, safety measures are taken, after which the miners can go down and work.’* (Female miner, FGD male and female miners, Kadoma, 2017) *‘That can be a belief* (of not using PPE) *among people* [ASG, miners], *but there should be order* (workplace organisation) *when people are at work. Anything can happen, but one should go down organised. People must not have an attitude to say, whatever happens, happens. If you do “chiite-ite” [“working haphazardly”], it’s risky, any accident can happen, and you get injured. We need to wear PPE, and we need a …. cloth, and we must wear it. Dust will not stop affecting you because you ..call.. yourself “Munija” [informal miner]. Dust will still get into your system, and over time you will get affected* (getting affected over time). *So, it’s important to work with protection. It’s important to teach each other to protect, not to destroy. Protection is important.’* (FGD, male miners, Kadoma, 2017) *‘In mining there must be avenues to get prompt results, like after taking water to the laboratory for analysis. Many questions and many charges will make people look for survival methods without taking the water for testing* (testing drinking water for chemical contamination e.g., cyanide, acids, mercury, caustic soda)*. People can look for shortcuts that can result in harmful effects in the future.’* (FGD, mine owners, Kadoma)
Behaviour	*‘The issue is* (that) *when hitting with the hammer, you hit hard and sweat, and it’s preferred for one to be free without PPE because one will be working hard.’ (FGD, male miners, Kadoma, 2017)* *‘When it was introduced* (the retort), *we did not take it up. But it’s* (the retort) *a good thing it recycles mercury…. The problem is we are used to our survival technics.’* (FGD, mine owners, Kadoma) *‘We will get behind time* (if we use the retort) *because our job in artisanal mining is seasonal; when rain begins we focus more on farming……. If the retort is slow we will put it aside, if the retort is time efficient then we will use it.’* (FGD, mine owners, Kadoma) *‘You get no gold without mercury.’* (FGD, male and female miners, Kadoma, 2017) *‘It depends on how the technology works. We are convinced after seeing how it works. At the moment we are convinced that if we use mercury we extract all the gold.’* (FGD, male and female miners, Kadoma, 2017)
Action and follow-up	*‘Every day before starting work, we have talks on the type of PPE one should use. Checking the PPE for snakes and scorpions before wearing PPE. There could be a snake or scorpion hiding in the gumboots; one must therefore check what could be hiding in the gumboots before wearing* (the gumboots)*.’* (FGD male and female miners, Kadoma, 2017) *‘Here, it’s a must to always wear PPE because if you are at work, you are expected to wear PPE until the end of your 8 hours.’* (FGD, male and female miners processing centre, 2017, Kadoma)

## Data Availability

Data are available at: Singo, Josephine (2022), “Survey on health and Safety in Artisanal & Small-Scale Gold Mining in Zimbabwe”, Mendeley Data, V1, doi:10.17632/v7wd3wjrxm.1. Data from in-depth interviews were collected from a different study and are available per request.

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
