# Peer review of "Health and Safety Risk Mitigation among Artisanal and Small-Scale Gold Miners in Zimbabwe"

_ijerph, 2022, doi:10.3390/ijerph192114352_

Round 1

Reviewer 1 Report

Review of A systematic approach to the causal analysis of accidents and 2 injuries among artisanal and small-scale gold miners in Zimbabwe.

Brief summary This manuscript aimed to describe a systematic approach to the causal analysis of accidents and injuries in artisanal and small-scale miners in Zimbabwe. The study objectives included investigating the causal factors of incidents in the ASGM in Zimbabwe using a systematic approach to causal analysis. The strengths seen in this paper are a clear concise description of findings and attribution of factors in ASGM to health and safety incidents. The main contribution is the detailed description of factors related to incidents in ASGM mining in Zimbabwe.

General concept Comments

The title refers to a systematic approach to the analysis yet this in not clear in the methods or discussion. The study appears to use common qualitative methods and no systematic method was described. The methodology does not come across clearly with the description of the study samples tied in with the methods used to collect data from the samples. The results are clearly presented but the discussion is lacking in detail and flow.

Introduction

Line 71-73 these two sentences are unclear; they refer to headlines (in newspapers? Online? Local or national?) and blaming the sector without preventative measures (clarify).

Line 78 Reason (date?),

Line 82 Bonsu (date?)

Line 87- 88 Clarify please, did the author develop the framework in fig 1? And why hence the relevance?

Methods

With two data collection time points perhaps a flow chart including the type of data collected and location will make understanding data collection easier. Some more details around participant selection are required such as how many mining sites/mines were selected and were the sites owned by the same or different owners how many owners? How many participants per site/mine in 2017 and 2020? Were mine owners only interviewed in 2017? Perhaps a separate study population paragraph would clarify things.

Line 120 Perhaps incident is a clearer term than phenomenon if that is what was meant or was it the situation?

Line 159 JB? Researcher?

Line 170 Either reference the multi-causal framework or provide it as supplementary information.

Line 175 JS?

Results

Table 1 Under Workplace factors, third row Blasting (4) appears 3 times.

Line 201 I do not see the relevance of the heading ASGM here.

Discussion

Line 484 and 486 discuss findings on beer and alcohol abuse not covered in the results

Line 507 to 517 is a duplicate of the paragraphs above, Lines 495- 504.

The discussion is short and just refers to literature as confirming findings or describes literature not relating to the findings. More interpretation of the results would strengthen the article. 

Author Response

Good day,

Thank you very much for the useful comments. Please find attached the responses to comments.

Please note that we have decided to make the article more comprehensive and conclude the study by:

  1. Upgrading the topic to risk mitigation thereby integrating causal analysis and risk mitigation
  2. Upgraded figure 1 to include risk mitigation
  3. Added more sections on health on results
  4. Added Figure 3 and Table 1
  5. Added health in the  discussion
  6. Added Supplementary figure 1.

My sincere apologies, I lost the margin lines in the process of editing.

Kind regards,

Josephine.

Reviewer 2 Report

It is a interesting topic. However, there are some things that should be sorted out before publication:

Despite the introduction has enough references and explains the lack of research and the main current lines, it does not flow well enough. I suggest the authors revise the whole section trying to improve the understanding, highlighting the lack of knowledge and the goals of the manuscript.

The first sentence “Mining is the most dangerous occupation” is inaccurate. Despite it has inherent risks there are other minor activities with riskier conditions. I think the sentence should be rephrased.

Figure 1 should be placed in the methodology, not the introduction section.

Rephrase line 117.

The footnotes should be integrated in the text.

Verify it is the last version, since there are some corrections in the document, such as page 9 or 10.

Future research lines should be included.

The conclusion section should be substantially improved, based on the results and achievement reached.

Author Response

Please note that we have decided to make the article more comprehensive and conclude the study by:

  1. Upgrading the topic to risk mitigation thereby integrating causal analysis and risk mitigation
  2. Upgraded figure 1 to include risk mitigation
  3. Added more sections on health on results
  4. Added Figure 3 and Table 1
  5. Added health in the  discusion
  6. Added Supplementary figure 1.
  7. My sincere apologies, I lost the margin lines in the process of edting

Kind regards,

Josephine.

Round 2

Reviewer 2 Report

I recommend you to attach a word file instead of a pdf, using a specific colour to highlight the changes instead of the changes control